# BioPETsurv: Methodology and open source software to evaluate biomarkers for prognostic enrichment of time-to-event clinical trials

Si Cheng[1], Kathleen F. Kerr[1], Heather Thiessen-Philbrook[2], Steven G. Coca[3], Chirag R. Parikh[2]*

1 Department of Biostatistics, University of Washington, Seattle, Washington, United States of America,
2 Division of Nephrology, Johns Hopkins University School of Medicine, Baltimore, Maryland, United States of America, 3 Division of Nephrology, Department of Medicine, Icahn School of Medicine at Mount Sinai, New York, New York, United States of America

* chirag.parikh@jhmi.edu

**Data Availability Statement:** The data underlying the results presented in the study are available from The Comprehensive R Network package

## Abstract

Biomarkers can be used to enrich a clinical trial for patients at higher risk for an outcome, a strategy termed "prognostic enrichment." Methodology is needed to evaluate biomarkers for prognostic enrichment of trials with time-to-event endpoints such as survival. Key considerations when considering prognostic enrichment include: clinical trial sample size; the number of patients one must screen to enroll the trial; and total patient screening costs and total per-patient trial costs. The Biomarker Prognostic Enrichment Tool for Survival Outcomes (BioPET-surv) is a suite of methods for estimating these elements to evaluate a prognostic enrichment biomarker and/or plan a prognostically enriched clinical trial with a time-to-event primary endpoint. BioPETsurv allows investigators to analyze data on a candidate biomarker and potentially censored survival times. Alternatively, BioPETsurv can simulate data to match a particular clinical setting. BioPETsurv's data simulator enables investigators to explore the potential utility of a prognostic enrichment biomarker for their clinical setting. Results demonstrate that both modestly prognostic and strongly prognostic biomarkers can improve trial metrics such as reducing sample size or trial costs. In addition to the quantitative analysis provided by BioPETsurv, investigators should consider the generalizability of trial results and evaluate the ethics of trial eligibility criteria. BioPETsurv is freely available as a package for the *R* statistical computing platform, and as a webtool at www.prognosticenrichment.com/surv.

## Introduction

Biomarkers are used for various purposes across research and clinical contexts. In a clinical trial of an intervention intended to prevent or delay some unwanted clinical event, a biomarker may be useful for "prognostic enrichment" of the trial [1–5]. A prognostically enriched trial uses a biomarker to enroll only patients at relatively higher risk of the unwanted clinical

BioPETsurv https://cran.r-project.org/web/packages/BioPETsurv/

**Funding:** Funding statement. NIH (R01-HL-085757 to CRP) funded the TRIBE-AKI Consortium. CRP is supported by NIH grant K24-DK090203 and the P30-DK-079310 O'Brien Kidney Center Grant. SGC has salary support from NIH grants R01 DK115562, U01 DK106962, R01 HL085757, R01 DK112258, and U01 OH011326. SGC and CRP are members of the NIH-sponsored Assessment, Serial Evaluation, and Subsequent Sequelae of Acute Kidney Injury (ASSESS-AKI) Consortium (U01-DK-082185). The funding agencies did not have any role in the study design, data collection and analysis, decision to publish, or preparation of the manuscript. The specific roles of CRP and SGC are articulated in the 'author contributions' section SGC received consulting fees from Goldfinch Bio, CHF Solutions, Quark Biopharma, Janssen Pharmaceuticals, Takeda Pharmaceuticals, and Relypsa. These organizations did not have any role in the study design, data collection and analysis, decision to publish, or preparation of the manuscript. The specific roles of SGC are articulated in the 'author contributions' section.

**Competing interests:** Competing Interests Statement SGC and CRP are members of the advisory board of RenalytixAI and own equity in the same. In the past 3 years, SGC has received consulting fees from Goldfinch Bio, CHF Solutions, Quark Biopharma, Janssen Pharmaceuticals, Takeda Pharmaceuticals, and Relypsa. All remaining authors have nothing to disclose. This does not alter our adherence to PLOS ONE policies on sharing data and materials.

event. Since study power depends on observing events, running a trial in an enriched study population can allow for a smaller trial compared to an unenriched trial [6, 7]. Moreover, it may be more ethically acceptable to test an intervention only on patients at high risk for the clinical event, and ethically preferable to test on a smaller study sample. Prognostic enrichment can produce greater efficiency in evaluating new interventions, potentially benefiting patients, sponsors, and public health.

There is a substantial literature on biomarkers that are predictive of treatment efficacy [8–12], also referred to as treatment-selection biomarkers [13–16]. In contrast, little has been written about evaluating biomarkers for prognostic enrichment [1, 2]. As noted by Temple [1], prognostic enrichment is well-established in cardiovascular disease, where it is common that interventions are first studied in individuals who are at high risk. CONSENSUS, the first trial of enalapril, enrolled only very high-risk patients (6-month mortality of 44%). CONSENSUS demonstrated efficacy of enalapril with only 253 patients. Subsequent trials in groups with less severe disease needed to be much larger.

In nephrology, a trial for patient with autosomal dominant polycystic kidney disease (ADPKD) enriched for those at greater risk of a substantial decline in renal function [17]. Total Kidney Volume (TKV), measured at baseline, was used in combination with patient age and estimated glomerular filtration rate (eGFR) to identify high risk patients. Without TKV, it was determined that 13 patients would need to be screened to enroll 11 patients to observe one event. With TKV, 25 patients would need to be screened to enroll 9 patients and observe one event. The FDA qualified TKV as a prognostic biomarker for use in clinical trials for ADPKD on August 31, 2015 [18]. The PRIORITY trial in patients with type 2 diabetes enriched for patients at high-risk of the primary endpoint, confirmed microalbuminuria, which occurred in 28% of participants classified as high-risk and only 9% of those classified as low-risk [19]. Although the trial did not establish that spironolactone is efficacious for the primary endpoint, without enrichment a sample size 3–4 times as large would have been needed and many more patients would have been exposed to a therapy that has side effects.

Despite prognostic enrichment being well-established in cardiology and employed in other clinical areas, little has been written about how to evaluate a biomarker for prognostic enrichment or to consider the trade-offs of an enriched vs. unenriched trial [4, 6]. For trials with a binary primary outcome, our group previously published the Biomarker Prognostic Enrichment Tool (BioPET) [7]. We identified key considerations for evaluating a biomarker for prognostic enrichment, including: clinical trial sample size; number of patients to screen to enroll the trial; total patient screening costs and the total of per-patient costs for patients in the trial. BioPET includes methods and graphical devices to evaluate a biomarker on these dimensions for trials with a binary outcome, but cannot be used for trials with a time-to-event outcome such as survival. Compared to trials with binary outcomes, trials with time-to-event outcomes can utilize more information in the data and accommodate the partial information available in censored outcomes. This article describes new methods and open source software, BioPET-surv, for such trials.

As a motivating example, consider the population of patients with a hospitalized episode of acute kidney injury [20] and a hypothetical intervention intended to prevent or delay the onset of chronic kidney disease. A randomized trial will compare the hazard for chronic kidney disease in a treatment group and a control group. As a proof-of-principle illustration, in this article we use synthetic data that mimic an existing cohort [20] to illustrate BioPETsurv for prognostic enrichment in this setting (Example 1).

BioPETsurv accommodates two trial designs. The first design is a fixed-duration trial – the observation period is the same for all patients. The second design has an accrual period plus a

follow-up period. For example, there may be a 1-year accrual period and a 3-year follow-up period, so that the observation period varies between 3 and 4 years for study participants.

BioPETsurv can be used to evaluate a biomarker and (possibly right-censored) survival data on a sample of patients. Alternatively, investigators can specify some essential features such as the event rate without enrichment and the prognostic capacity of the biomarker in terms of a hazard ratio. BioPETsurv can simulate biomarker and survival data matching these specifications, allowing investigators to explore prognostic enrichment for their clinical setting.

In this article, "biomarker" can refer to either a single measured characteristic or a "composite biomarker" [2] combining multiple prognostic factors [7]. For simplicity, we use "survival" for any time-to-event variable.

## Methods

Without loss of generality, assume that patients with higher levels of the biomarker tend to experience the unwanted clinical outcome sooner. For a binary outcome, the area under the ROC curve (AUC) summarizes the discrimination performance of a biomarker. For a survival outcome, BioPETsurv displays the Kaplan-Meier survival curves for the entire patient population and for enriched subsets.

A continuous biomarker can, in principle, be used for a low or high level of enrichment of a trial. The level of enrichment is the threshold (percentile in the biomarker) above which patients are eligible for the trial. For example, excluding patients from the trial below the 10[th] percentile in the biomarker would be a low level of enrichment; requiring patients in the trial to be in the top quartile would be a high level of enrichment. Based on the level of enrichment, the prognostic strength of the biomarker, and the length of the trial, BioPETsurv estimates the expected **event rate** absent intervention. The expected event rate together with statistical testing specifications (e.g., power) and the treatment effect to detect determine the **trial sample size.** The **total number of patients screened** to enroll the trial depends on the trial sample size and the level of enrichment. For example, a trial with a 50% level of enrichment requires, on average, 2 patients to be evaluated to identify one eligible for the trial. Under the assumption that patients express interest in enrolling in the trial at a constant rate over time, 'total number of patients screened' is a proxy for the calendar time to enroll the trial [7].

For cost analysis, BioPETsurv allows the cost for a patient in the trial to be either constant, or depend on the time the patient is in the trial before the primary endpoint. The latter may be realistic if the endpoint is death. The cost of screening, such as assay costs or patient work-up, must also be specified. Based on these user-specified costs, BioPETsurv calculates total trial cost for each enrichment level.

A key element in prognostic enrichment is the time-specific event rate by the end of the trial in enriched subgroups, which must be estimated. This can be done using Kaplan-Meier methods in subgroups. Alternatively, the nearest neighbor estimation method [4] allows the censoring process to depend on the biomarker and guarantees monotone estimated Receiver Operating Characteristic curves for time-specified outcomes. BioPETsurv offers both methods for fixed-duration trials and uses Kaplan-Meier methods for trials with an accrual period plus a follow-up period.

### Fixed-duration trials

Given type I error rate α, power 1-β, and treatment hazard ratio HR, the number of events

needed [21] is $N_0 = \frac{4\left(z_{1-\alpha/2}+z_{1-\beta}\right)^2}{\log^2 \mathrm{HR}}$. For a given enrichment level and trial length, let $\hat{S}$ be

estimated survival; the event rate is $\hat{p}_C = 1 - \hat{S}$ in the control arm and $\hat{p}_T = 1 - \hat{S}^{HR}$ in the treatment arm. Let $N_{\frac{1}{2}}$ be the sample size in one arm of a trial planned to have equal sample size in each arm. Then $N_{\frac{1}{2}} \times (\hat{p}_C + \hat{p}_T) = N_0$, so total $N$ is $2N_{\frac{1}{2}} = {}^2N_0 / (\hat{p}_C + \hat{p}_T)$. Let $C_1$ be the cost for a patient in the trial and $C_2$ the cost of screening (such as assay cost). For enrichment at quantile t (patients with biomarker below quantile t are excluded), total cost is $C_1 N + C_2 \frac{N}{1-t}$. Let $\hat{p} = \hat{p}_C + \hat{p}_T$. We calculate the standard deviation (SD) from the delta method, $SD(\hat{p}) = SD(2 - \hat{S} - \hat{S}^{HR}) \approx [1 + HR \cdot \hat{S}^{HR-1}]SD(\hat{S})$ and $SD(N) = SD\left(\frac{2N_0}{\hat{p}}\right) \approx \frac{2N_0}{\hat{p}^2} SD(\hat{p})$. We treat $N_o$, which comes from a standard sample size formula, as fixed; variability comes from $\hat{p}$.

## Trials with an accrual period and a follow-up period

Let $a$ and $f$ be the accrual and follow-up time respectively. The only difference from a fixed-duration trial is in estimating the event rates, $\hat{p}_C$ and $\hat{p}_T$, when participants are followed for different periods of time. Following [21], $\hat{p}_C = 1 - \frac{1}{6}[\hat{S}(f) + 4\hat{S}(f + 0.5a) + \hat{S}(f + a)]$ from Simpson's rule and $\hat{p}_T = 1 - \frac{1}{6}[\hat{S}(f)^{HR} + 4\hat{S}(f + 0.5a)^{HR} + \hat{S}(f + a)^{HR}]$, with standard errors estimated by bootstrapping. All other trial characteristics follow as for fixed-duration trials.

## Simulating data to allow investigators to explore prognostic enrichment

To allow investigators to explore prognostic enrichment without data on survival and the biomarker for a sample of patients, BioPETsurv can simulate data to mimic specific clinical parameters. Survival is simulated from a Weibull distribution with user-specified shape parameter $k$, which allows hazards to be constant, increasing, or decreasing. The user also specifies the survival probability $p$ at time T, which are used to solve for the Weibull scale parameter a. We expect investigators will specify $p$ based on knowledge of overall survival in the patient population. The data simulator takes $p$ as the survival probability for individuals with mean biomarker level. The survival curve for this group is similar to the overall survival curve. The prognostic strength of the biomarker is specified by the hazard ratio for a 1 standard deviation difference in the biomarker. Without loss of generality, the biomarker X is mean-centered so that X = 0 is the baseline group. Given Weibull shape and scale parameters, baseline hazards are $\lambda_0(t) = \frac{k}{a}\left(\frac{t}{a}\right)^{k-1}$ and under proportional hazards an individual with biomarker X has hazard $\lambda(t) = \frac{k}{a}\left(\frac{t}{a}\right)^{k-1}e^{\beta X}$, which corresponds to a Weibull distribution with the same shape parameter k and scale parameter $a \cdot e^{-\frac{\beta X}{k}}$. Biomarker data are simulated to have either a symmetric (normal) or right-skewed (lognormal) shape (user-specified). Based on biomarker value X = x, survival time is simulated from the appropriate Weibull distribution but censored at time T. The joint distribution of simulated biomarker and survival times is used for prognostic enrichment analysis.

## Results

### Example 1: A modestly prognostic biomarker and fixed-duration trial

Fig 1 and Table 1 show an example using BioPETsurv to evaluate a biomarker that is modestly prognostic of the event, with HR 1.2 corresponding to one standard deviation difference in the biomarker. The trial will be either 36 or 48 months. Fig 1A shows estimated survival curves for screening threshold 0% (top curve), i.e., for all patients (no enrichment). The plot shows that events accumulate more quickly in enriched subpopulations of patients, showing more quickly decreasing survival curves for enrichment levels 25%, 50%, and 75% (meaning that patients with biomarker below the 25th, 50th, or 75th percentile are excluded). Two vertical lines indicate the

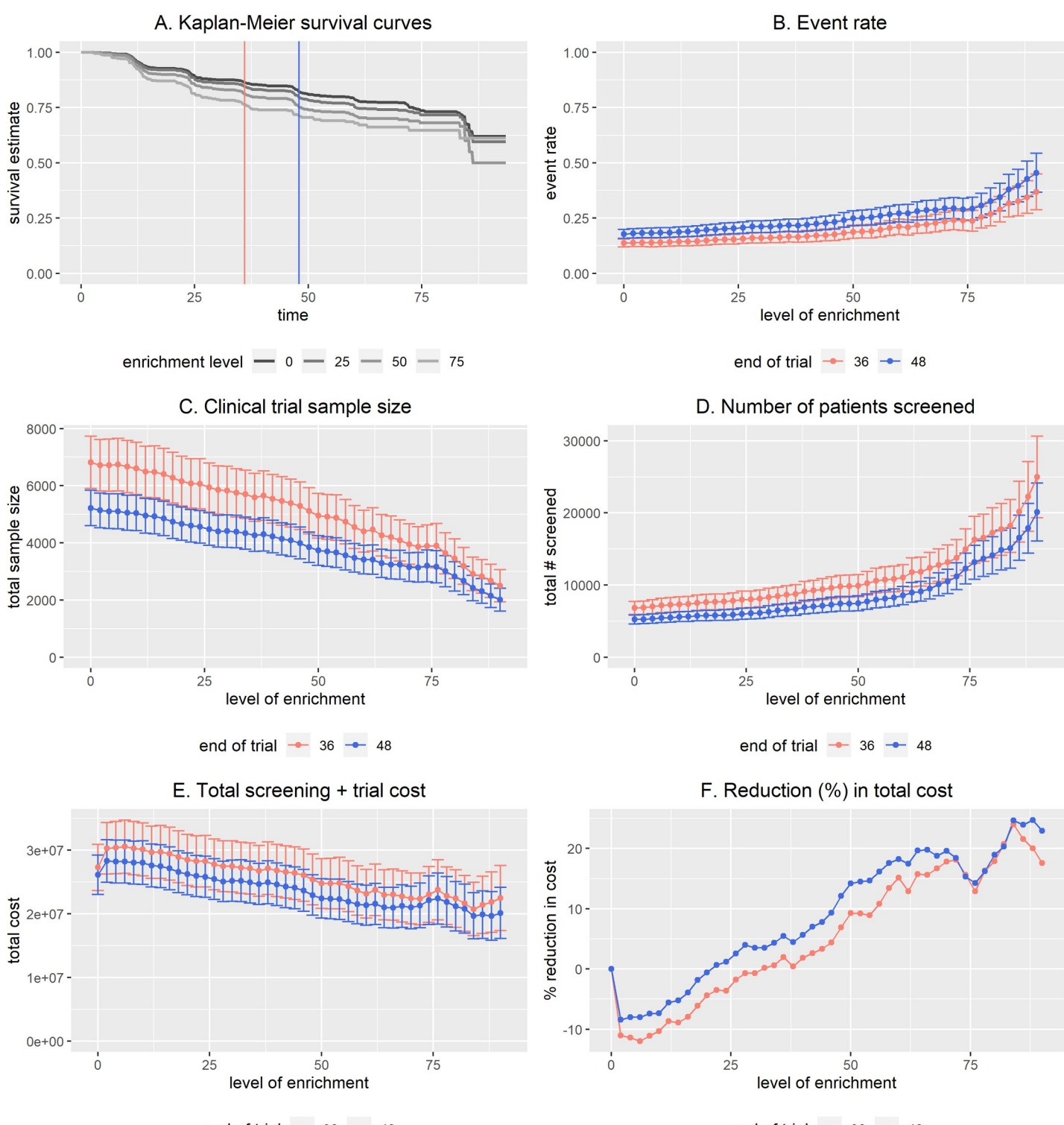

**Fig 1. BioPETsurv analysis of a modestly prognostic biomarker for a fixed-duration 36-month or 48-month trial.** Investigators are considering the biomarker for enrichment of either a 36-month or 48-month trial and specified 90% power to detect a hazard ratio of 0.8 using two-sided testing and α = 0.05. For cost analysis, the cost of screening was $500 and the cost of one patient in the trial was $4000 (36-month trial) and $5000 (48-month trial). The biomarker in this example has HR≈1.2 corresponding to a 1 SD difference in the marker.

**Table 1. BioPETsurv analysis of a modestly prognostic biomarker (Example 1).**

| Screening Threshold | Event Rate (%) | | Sample Size | | Total Screened | | Reduction in Total Cost (%) | |
|---|---|---|---|---|---|---|---|---|
| | 36 mo | 48 mo | 36 mo | 48 mo | 36 mo | 48 mo | 36 mo | 48 mo |
| 0% | 14 | 18 | 6819 | 5221 | 6819 | 5221 | 0 | 0 |
| 5% | 14 | 18 | 6726 | 5102 | 7080 | 5371 | -12 | -8 |
| 10% | 14 | 18 | 6605 | 5045 | 7339 | 5606 | -10 | -7 |
| 15% | 14 | 19 | 6441 | 4879 | 7578 | 5740 | -8 | -4 |
| 20% | 15 | 20 | 6155 | 4668 | 7694 | 5835 | -4 | -1 |
| 25% | 15 | 20 | 6014 | 4542 | 8019 | 6056 | -3 | 1 |
| 30% | 16 | 21 | 5825 | 4408 | 8322 | 6298 | -1 | 4 |
| 35% | 16 | 22 | 5643 | 4317 | 8682 | 6642 | 1 | 5 |
| 40% | 17 | 22 | 5540 | 4221 | 9234 | 7035 | 2 | 6 |
| 45% | 18 | 23 | 5299 | 4013 | 9635 | 7297 | 5 | 9 |
| 50% | 19 | 25 | 4949 | 3733 | 9898 | 7466 | 9 | 14 |
| 55% | 19 | 25 | 4845 | 3648 | 10767 | 8107 | 9 | 15 |
| 60% | 21 | 27 | 4407 | 3414 | 11018 | 8536 | 15 | 18 |
| 65% | 22 | 29 | 4141 | 3189 | 11832 | 9112 | 18 | 21 |
| 70% | 23 | 29 | 3956 | 3147 | 13187 | 10491 | 18 | 20 |
| 75% | 24 | 29 | 3905 | 3179 | 15620 | 12716 | 14 | 15 |
| 80% | 27 | 33 | 3445 | 2821 | 17226 | 14106 | 18 | 19 |
| 85% | 32 | 39 | 2880 | 2373 | 19201 | 15821 | 23 | 24 |
| 90% | 37 | 45 | 2497 | 2012 | 24971 | 20121 | 18 | 23 |

In 36 months the clinical event occurs in 13% +/- 1% of patients without intervention; and 18% +/- 1% in 48 months. Sample size calculations reflect 90% power to detect 0.8 treatment hazard ratio using two-sided hypothesis testing and $\alpha = 0.05$. The cost of screening is $500/patient and the per patient trial cost is $4000 (36-month trial) or $5000 (48-month trial). **Screening Threshold** is the proportion of patients who will be screened out of the trial based on biomarker level. **Event Rate** is the estimated rate of the clinical event in the enriched study population not receiving the intervention. **Sample Size** is the trial sample size calculated based on the event rate and statistical testing specifications. **Total Screened** is the total number of patients who would need to be screened to enroll the trial, which depends on the sample size and level of enrichment (screening threshold). **Total Cost** summarizes patient-related costs of different levels of enrichment, specifically the cost of biomarker-based screening and the cost of having a patient in a trial. Results show the potential for the biomarker to allow substantially smaller trial sample size and cost savings, but impose a greater burden on the total number of patients to screen to enroll the trial. These results are displayed in Fig 1, which also displays standard error estimates.

candidate trial lengths, 36 and 48 months. In all other panels in Fig 1, the horizontal axis is the level of enrichment, with 0% representing an unenriched trial. Fig 1B shows the estimated event rate as a function of the level of enrichment for both candidate trial lengths. Based on these event rates and specifying 90% power to detect treatment hazard ratio 0.8 (two-sided testing, $\alpha = 0.05$), Fig 1C shows the sample size for each trial duration. As expected, the longer trial requires fewer patients than the shorter trial. Fig 1D shows the number of patients needed to screen to enroll the trial. With this modestly prognostic biomarker, the screening total increases with higher enrichment, although the increase is modest below 50% enrichment. Fig 1E and 1F display the cost analysis, with per-patient costs of $4000 (36-month trial) and $5000 (48-month trial). The screening cost was set at $500. In this example, with less than 25% enrichment an enriched trial is anticipated to be more expensive than an unenriched trial because the decrease in sample size is not enough to offset the cost of screening. The cost analysis shows cost savings for higher levels of enrichment.

## Example 2: Simulating data for a highly prognostic biomarker and a trial with accrual period and follow-up period

We illustrate the BioPETsurv data simulator. We set simulation parameters to mimic the clinical setting of Example 1 but anticipating a more highly prognostic biomarker. We simulated

biomarker and survival data for n = 5,000 patients with event rate 18% at 48 months. We specified constant hazards, and a hazard ratio 2.8 corresponding to 1 standard deviation difference in the biomarker, which we simulated as normally distributed. Fig 2 and Table 2 give prognostic enrichment analysis using the simulated data and planning a trial with a 12-month accrual period and a 36-month follow-up. We again specified 90% power to detect 0.8 treatment hazard ratio, two-sided testing, and α = 0.05.

Fig 2A shows estimated survival curves for no enrichment, and 25%, 50%, and 75% enrichment. Compared to Fig 1A, there is greater separation between the curves because the biomarker here is more prognostic. Fig 2B shows the average event rate for each level of enrichment (the average accounts for the variable length of patient follow-up). Sample size decreases steadily with greater enrichment (Fig 2C). The total number of patients to screen to enroll the trial is gradually increasing for lower levels of enrichment but increases rapidly at high levels of enrichment (Fig 2D). With $300 screening cost and a patient in the trial costing $100/month before the clinical event, there is potential for substantial savings from prognostic enrichment (Fig 2E and 2F).

## Discussion

In this work we demonstrated a comprehensive framework for evaluating a biomarker for prognostic enrichment of a clinical trial with a survival endpoint. In both Examples 1 and 2, total trial costs are nearly monotone decreasing with greater levels of enrichment, but this will not always be true. For example, one can use the data simulator with the following specifications: $100 screening cost, $1000 cost per patient in the trial, 50% survival at 10 years and a trial planned for 5 years for 90% power to detect a treatment hazard ratio of 0.8. If the biomarker is highly prognostic (effect size 2.0), the total trial cost is U-shaped with a minimum at about the 75% enrichment level (that is, the trial only enrolls patients in the highest quartile of the biomarker). See S1 Fig and S1 Table. On the other hand, If the biomarker is weakly prognostic (e.g., effect size 1.2), total cost is monotone increasing with the level of enrichment. See S2 Fig and S2 Table. That is, at a 1:10 ratio of per patient screening and trial costs, it is not cost-effective to use prognostic enrichment at any level with the weakly prognostic biomarker.

Interestingly, the number of patients screened can be either an increasing or decreasing function of the level of enrichment. In both Examples 1 and 2 it was increasing. However, for highly prognostic biomarkers, the number of patients screened can be decreasing because the trial sample size drops precipitously and more than compensates for the additional patients who must be screened for an enriched trial (see [7] for examples).

The BioPETsurv data simulator requires specification of the biomarker distribution, the biomarker hazard ratio, the trend in hazards over time (increasing, constant, or decreasing), and the event rate without enrichment. These elements are realistic for area specialists to identify. The simulation methodology incorporates proportional hazards. As with any data simulation, results will be accurate only to the extent that the assumptions of the simulation hold.

When considering a prognostic enrichment strategy, investigators must consider multiple, sometimes conflicting goals: trial sample size, number of patients to screen for eligibility, and cost. BioPETsurv is useful for several types of questions in this arena. First, investigators with data on a prognostic biomarker can use BioPETsurv to evaluate that biomarker for its utility for prognostic enrichment for their clinical setting. Second, investigators with a prognostic biomarker who are planning a trial can use BioPETsurv to decide whether, and to what extent, to use the biomarker to plan and implement an enriched trial. Third, investigators working in a particular clinical setting can use BioPETsurv's simulation functionality to explore the prognostic capacity that would be needed in order for a biomarker to be useful for prognostic

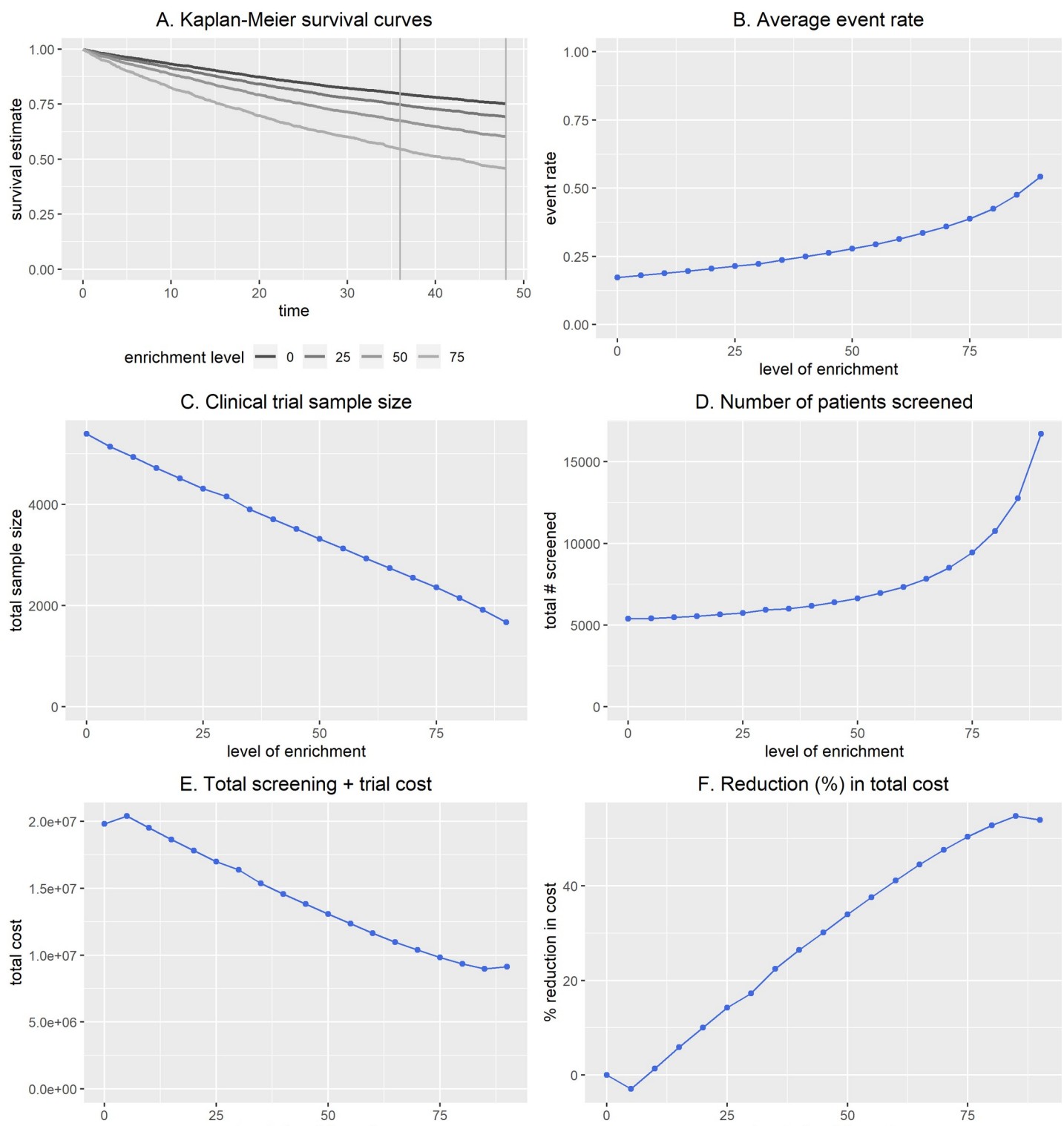

**Fig 2. BioPETsurv analysis of simulated biomarker for a trial with a 12-month accrual period and 36-month follow-up period.** Investigators are planning a trial with a 12-month accrual period plus a 36-month follow-up period, and anticipate having a marker with HR≈2.8 corresponding to a 1 standard deviation difference in the marker. The BioPETsurv data simulator generated data for a normally distributed biomarker with this prognostic strength. Sample size calculations specified 90% power to detect a treatment hazard ratio of 0.8 using two-sided testing and α = 0.05. For cost analysis, patient screening cost was $300 and the cost of a patient in a trial was $100/month before the clinical endpoint. Numeric results are in Table 2.

**Table 2. BioPETsurv analysis of simulated biomarker for a trial with a 12-month accrual period and 36-month follow-up period (Example 2).**

| Screening Threshold | Event Rate (%) | Sample Size | Total Screened | Reduction in Total Cost (%) |
|---|---|---|---|---|
| 0% | 17 | 5394 | 5394 | 0 |
| 5% | 18 | 5146 | 5417 | -3 |
| 10% | 19 | 4937 | 5486 | 1 |
| 15% | 20 | 4720 | 5553 | 6 |
| 20% | 21 | 4517 | 5647 | 10 |
| 25% | 22 | 4311 | 5748 | 14 |
| 30% | 22 | 4155 | 5936 | 17 |
| 35% | 24 | 3905 | 6008 | 22 |
| 40% | 25 | 3706 | 6177 | 26 |
| 45% | 26 | 3518 | 6397 | 30 |
| 50% | 28 | 3320 | 6640 | 34 |
| 55% | 29 | 3131 | 6958 | 38 |
| 60% | 31 | 2935 | 7338 | 41 |
| 65% | 34 | 2741 | 7832 | 45 |
| 70% | 36 | 2553 | 8511 | 48 |
| 75% | 39 | 2360 | 9440 | 50 |
| 80% | 43 | 2153 | 10766 | 53 |
| 85% | 48 | 1916 | 12774 | 55 |
| 90% | 54 | 1670 | 16701 | 54 |

Investigators are planning a trial with a 12-month accrucal period plus a 36-month follow-up period, and anticipate having a biomarker with HR≈2.8 corresponding to a 1 standard devfiation difference in the marker. The BioPETsurv data simulator generated data for a normally distributed biomarker with this prognostic strength. Sample size calculatations specified 90% power to detect a treatment hazard ratio of 0.8 using two-sided testing and α = 0.05. For cost analysis, patient screening cost was set to $300 and the cost of a patient in a trial was set to $100/month before the clinical endpoint. Results are displayed in Fig 2.

enrichment; results then inform biomarker research and development in that area [22–25]. BioPETsurv uses metrics that align with key trial elements. Together with important considerations around generalizability and ethics, BioPETsurv facilitates a comprehensive evaluation of competing dimensions in trial planning and the evaluation of prognostic enrichment biomarkers.

## Supporting information

**S1 Fig.**
(TIF)

**S2 Fig.**
(TIF)

**S1 Table.**
(DOCX)

**S2 Table.**
(DOCX)

## Author Contributions

**Conceptualization:** Kathleen F. Kerr, Chirag R. Parikh.

**Data curation:** Heather Thiessen-Philbrook.

**Formal analysis:** Si Cheng.

**Funding acquisition:** Chirag R. Parikh.

**Methodology:** Si Cheng, Kathleen F. Kerr.

**Project administration:** Heather Thiessen-Philbrook.

**Software:** Si Cheng.

**Supervision:** Kathleen F. Kerr.

**Visualization:** Si Cheng.

**Writing – original draft:** Kathleen F. Kerr.

**Writing – review & editing:** Si Cheng, Heather Thiessen-Philbrook, Steven G. Coca, Chirag R. Parikh.

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
