## [Decision Letter · Decision Letter 0]

19 Jun 2020

PONE-D-20-12411

BioPETsurv:  Methodology and open source softwre to evaluate biomarkers for prognostic enrichment of time-to-event clinical trials

PLOS ONE

Dear Dr. Parikh,

Thank you for submitting your manuscript to PLOS ONE. After careful consideration, we feel that it has merit but does not fully meet PLOS ONE’s publication criteria as it currently stands. Therefore, we invite you to submit a revised version of the manuscript that addresses the points raised during the review process.

We look forward to receiving your revised manuscript.

Kind regards,

Qin Liu

Academic Editor

PLOS ONE

Journal Requirements:

"SGC and CRP are members of the advisory board of RenalytixAI and own equity in the same. In the past 3 years, SGC has received consulting fees from Goldfinch Bio, CHF Solutions, Quark Biopharma, Janssen Pharmaceuticals, Takeda Pharmaceuticals, and Relypsa."

We note that one or more of the authors are employed by a commercial company: RenalytixAI, Goldfinch Bio, CHF Solutions, Quark Biopharma, Janssen Pharmaceuticals, Takeda Pharmaceuticals, and Relypsa.

2.1. Please provide an amended Funding Statement declaring this commercial affiliation, as well as a statement regarding the Role of Funders in your study. If the funding organization did not play a role in the study design, data collection and analysis, decision to publish, or preparation of the manuscript and only provided financial support in the form of authors' salaries and/or research materials, please review your statements relating to the author contributions, and ensure you have specifically and accurately indicated the role(s) that these authors had in your study. You can update author roles in the Author Contributions section of the online submission form.

2.2. Please also provide an updated Competing Interests Statement declaring this commercial affiliation along with any other relevant declarations relating to employment, consultancy, patents, products in development, or marketed products, etc. 

Reviewers' comments:

Reviewer's Responses to Questions

**Comments to the Author**

1. Is the manuscript technically sound, and do the data support the conclusions?

Reviewer #1: Yes

Reviewer #2: Partly

2. Has the statistical analysis been performed appropriately and rigorously? 

Reviewer #1: Yes

Reviewer #2: N/A

3. Have the authors made all data underlying the findings in their manuscript fully available?

Reviewer #1: Yes

Reviewer #2: Yes

4. Is the manuscript presented in an intelligible fashion and written in standard English?

Reviewer #1: Yes

Reviewer #2: No

5. Review Comments to the Author

Reviewer #1: Please see attached.

Please see attached.

Please see attached.

Please see attached.

Please see attached.

Please see attached.

Please see attached.

Please see attached.

Please see attached.

Please see attached.

Reviewer #2: The study introduced a methodology and software called BioPETsurv. The work allows the end-users to plan for a trial with prognostic enrichment, in which the primary endpoint is time-to-event. Through simulation, some key operating characteristics, such as the number of patients to screen for eligibility and trial cost, can be generated to assist the trial planning. The free software in an R package: BioPETsurv will be helpful for many end-users. However, the overall scientific presentation should be improved to assist in comprehending. A few suggestions are listed below

1. Introduction

a. Line 46-56: Elaborate more about the history, concept, and definition of prognostic enrichment or a prognostically enriched trail by providing real-life examples.

b. Line 57-60: Elaborate what the predictive biomarker is and add references; why evaluating biomarkers for prognostic enrichment is of importance; When you say ‘sensible analyses,’ do you mean ‘sensitivity analyses’? If not, please clarify.

c. Line 61-64: Elaborate more about why those considerations would be meaningful in trial designs.

d. Line 71-76: for the motivating example, please provide more clinical background and design considerations in the trial (reference #8). Did the trial stratify by predictive biomarker? How did a prognostic enrichment design implement? For what purpose?

e. Line 82-83: BioPETsurv is a tool to help design a trial that aims to evaluate a predictive biomarker with a time-to-event primary endpoint, or BioPETsurv itself can help evaluate biomarkers?

f. Line 88-90: Should the biomarker here be a prognostics biomarker or a predictive biomarker? The two definitions are different, and there are many works of literature to delineate them.

2. Method section

a. Line 93-95: consider revising as ‘For a binary outcome, the area under the curve (AUC) by the ROC analysis summarizes the performance of discrimination of a biomarker.’

b. Line 95-97: KM curves for the time-to-event outcome are not analogous to the ROC curve for a binary outcome. However, for a time-to-event outcome, the researcher can generate time-specific AUC (PMID: 28388943)

c. Line 98: what is the definition of the level of enrichment?

d. Line 105: Not sure what the sentence means? Please revise.

e. Line 112: should it be ‘time-specific event rate’?

f. Line 121: what is the definition of the treatment hazard ratio? The HR for treated arm vs. control arm with enrichment level = 0? Or is it the HR corresponding to one standard deviation difference in the biomarker?

g. Line 129: what is the interpretation of the function of SD?

h. Line 146: in practice, how will time T be decided?

i. Line149-150: the sentence (‘but we have seen that ….’) is hard to understand, please revise.

3. Results section

a. It will be helpful to illustrate the example in the context of motivating case study (reference #8). What is the objective/purpose of design to including enrichment consideration? Why were the parameters of simulation set up at a specific value? What will preliminary data be essential for the simulation to be useful?

b. Figures 1 and 2, it is helpful to show the pattern change by the level of enrichment, but the clinical trial sample size and the number of patients screened could be too high to be realistic.

c. In a real trial design, how to reasonably determine those parameters for simulation?

d. I hope these tutorial examples can help future users to design their next trial.

4. Discussion Section

a. What is the novelty of the proposed study? Under what circumstance, the end-users should consider using the proposed approach.

b. Any other similar studies, if yes, what are the advantages and disadvantages of the current one.

c. Any other considerations the end-users should be aware of in order to generate the simulation that closely mimics the reality.

6. PLOS authors have the option to publish the peer review history of their article (what does this mean?). If published, this will include your full peer review and any attached files.

Reviewer #1: Yes: Jiangtao Gou

Reviewer #2: No

---

## [Author Response · Author response to Decision Letter 0]

10 Jul 2020

Reviewer #1

Authors developed an R package BioPETsurv (released in January 2020) and an R shiny app available at http://prognosticenrichment.com/surv/ as a biomarker prognostic enrichment tool for survival analysis. The R package BioPETsurv is an extension of another package BioPET (released in July 2018). With these tools, clinical trialists can develop a protocol on a trial with prognostic enrichment faster and more effectively. This package also allows investigators to explore prognostic enrichment with simulated data. The total cost is computed as a linear combination of the cost for a patient in the trial and that of screening. The sample size estimations for fixed-duration trials and trials with an accrual period and a follow-up period follow the formulas in Schoenfeld (1983, Biometrics), Sample-size formula for the proportional-hazards regression model. This manuscript is well-structured and well written, and the documents for the R package and shiny app are also well-written. I only have a few comments below.

Comments

1. Under the settings in Table 1 and 2, the Reduction in Total Cost is generally an increasing function of the Screening Threshold. Authors can add additional situation where the Reduction in Total Cost is not a monotonic function of the Screening Threshold. For example, with effect size 2, Cost of screening a patient to determine trial eligibility 100 and Cost of running a patient through the trial 100, the maximum Reduction in Total Cost is achieved around Screening

Threshold = 75%.

Response: Thank you. We have added this example to the Discussion, where we have also added additional material of the different types of results one can see with trial metrics. In the interest of keeping our paper succinct, encouraging readers to explore our webtool, and to avoid repetition with material in Kerr et al (Clinical Trials, 2017), we did not add figures and tables for these examples to this paper. Readers can generate results themselves using our webtool. However, we would be happy to extend our paper to add more extensive results on additional examples at the direction of the editors.

2. Authors can discuss the general relation between the Reduction in Total Cost and the Screening Threshold. For example, when the relation is monotonic, when it is not.

Response: Thank you. As mentioned above, we have extended the Discussion section to describe the behavior of trial metrics.

3. The resolution of Figure 1 and 2 is quite low. I guess the reason is that the ggplot figures were first imbedded into MS Word and then were converted to pdf. Please make sure to have high resolution images for the final submission.

Response: Thank you. We have uploaded higher resolution figures with the revision.

4. There is a typo on the shiny app: “eligiblity” should be spelled as “eligibility”.

Response: Thank you very much for alerting us of this typo, which we have corrected.

Reviewer #2: The study introduced a methodology and software called BioPETsurv. The work allows the end-users to plan for a trial with prognostic enrichment, in which the primary endpoint is time-to-event. Through simulation, some key operating characteristics, such as the number of patients to screen for eligibility and trial cost, can be generated to assist the trial planning. The free software in an R package: BioPETsurv will be helpful for many end-users. However, the overall scientific presentation should be improved to assist in comprehending. A few suggestions are listed below

1. Introduction

a. Line 46-56: Elaborate more about the history, concept, and definition of prognostic enrichment or a prognostically enriched trail by providing real-life examples.

Response: Thank you. In the revised article we describe the CONSENSUS trial, which Temple (2010) considers a classic example of a trial using prognostic enrichment, and two more recent examples from nephrology.

b. Line 57-60: Elaborate what the predictive biomarker is and add references; why evaluating biomarkers for prognostic enrichment is of importance; When you say ‘sensible analyses,’ do you mean ‘sensitivity analyses’? If not, please clarify.

Response: Thank you. We have elaborated on that predictive biomarkers are useful for treatment selection because, by definition, they predict treatment benefit. We added 9 references on predictive biomarkers.

We meant "sensible analyses" and not "sensitivity analyses" – in using this term we intended to acknowledge that some of the ideas presented in our paper have appeared elsewhere. Our methodology brings these principles together in a unified, comprehensive framework, and our open source software facilitates the systematic application of these principles in practice. For clarity we have removed the term "sensible analyses" and re-written the relevant section of the Introduction. 

c. Line 61-64: Elaborate more about why those considerations would be meaningful in trial designs.

Response: We believe most readers of this article will have be familiar with the challenges and constraints in designing a clinical trial. Due to the expense and difficulty of recruiting patients to a trial as well as the ethical preference to experiment on fewer rather than more patients, we believe it is clear that trialists are keenly interested in minimizing the sample size needed for a trial they are planning. In the interest of brevity, we have not expanded this sentence as the reviewer suggests, but would be happy to do so if directed by the editor.

d. Line 71-76: for the motivating example, please provide more clinical background and design considerations in the trial (reference #8). Did the trial stratify by predictive biomarker? How did a prognostic enrichment design implement? For what purpose?

Response: The reference (reference #8 in the first submission) is an observational study. There is general interest in trials of interventions to help prevent chronic kidney disease (CKD), but CKD is sufficiently uncommon such that large sample sizes would be required to run an adequately powered trial. Therefore, this is an example where a prognostic enrichment strategy could make a clinical trial more efficient and less expensive by testing treatment efficacy using fewer trial subjects (which is ethically preferable in itself). The reference does not describe an actual trial so we are unable to make the changes that the reviewer describes. In the revision, we have added the word "hypothetical" to make clear that we do not describe an actual trial.

e. Line 82-83: BioPETsurv is a tool to help design a trial that aims to evaluate a predictive biomarker with a time-to-event primary endpoint, or BioPETsurv itself can help evaluate biomarkers?

Response: Thank you. The reviewer is correct that BioPETsurv can be used for different precise purposes:

1. Investigators with data on a prognostic biomarker can use BioPETsurv to evaluate that biomarker for its utility for prognostic enrichment for their clinical setting.

2. Investigators with a prognostic biomarker who are planning a trial can use BioPETsurv to decide whether, and to what extent, to use the biomarker to plan and run an enriched trial.

3. Investigators working in a particular clinical setting can ask the question: what prognostic capacity would I need in order for a biomarker to be useful for prognostic enrichment? BioPETsurv's simulation functionality allows investigators to explore this question.

The Discussion of the revised submission is extensively revised, and better identifies and delineates these related, but distinct, goals. We are grateful to the reviewer for noting the ambiguity in our original submission.

f. Line 88-90: Should the biomarker here be a prognostics biomarker or a predictive biomarker? The two definitions are different, and there are many works of literature to delineate them.

Response: We have changed "predictors" to "prognostic factors" in this sentence to avoid confusion between prognostic and predictive biomarkers.

2. Method section

a. Line 93-95: consider revising as ‘For a binary outcome, the area under the curve (AUC) by the ROC analysis summarizes the performance of discrimination of a biomarker.’

Response: We have revised the sentence similar to the reviewer's suggestion.

b. Line 95-97: KM curves for the time-to-event outcome are not analogous to the ROC curve for a binary outcome. However, for a time-to-event outcome, the researcher can generate time-specific AUC (PMID: 28388943)

Response: Thank you. We agree with the reviewer that KM curves are not truly analogous to ROC curves. However, KM curves are able to communicate the prognostic capacity of a biomarker across a continuous range of time points. We have revised this sentence to remove the misleading statement that KM curves are analogous to ROC curves.

c. Line 98: what is the definition of the level of enrichment?

Response: Thank you. We have added the following sentences to make this clear: " A continuous biomarker can, in principle, be used for a low or high level of enrichment of a trial. The level of enrichment is the threshold (percentile in the biomarker) above which patients are eligible for the trial. Excluding patients from the trial below the 10th percentile in the biomarker would be a low level of enrichment; requiring patients in the trial to be in the top quartile would be a high level of enrichment."

d. Line 105: Not sure what the sentence means? Please revise.

Response: Thank you. This sentence refers to the "total number of patients screened to enroll the trial", which is described in the immediately preceding sentence. We have revised the confusing sentence to state: "Under the assumption that patients express interest in enrolling in the trial at a constant rate over time, `total number of patients screened' is a proxy for the calendar time to enroll the trial".

e. Line 112: should it be ‘time-specific event rate’?

Response: Yes, thank you. We have made this revision.

f. Line 121: what is the definition of the treatment hazard ratio? The HR for treated arm vs. control arm with enrichment level = 0? Or is it the HR corresponding to one standard deviation difference in the biomarker?

Response: The treatment hazard ratio (HR) is the treated arm vs. control arm. Since the biomarker is prognostic and not predictive, the HR is the same regardless of enrichment level – it is unnecessary to specify enrichment level = 0.

g. Line 129: what is the interpretation of the function of SD?

Response: It is the standard deviation. We have clarified this in the text.

h. Line 146: in practice, how will time T be decided?

Response: In practice, T will be decided according to the application. For example, if a treatment is intended to prevent relapse for cancer patients in remission, T will be shorter for cancers where relapse normally occurs soon after remission and T will be longer for cancers where relapse is typically more delayed.

i. Line149-150: the sentence (‘but we have seen that ….’) is hard to understand, please revise.

Response: Thank you. The paper now states: " The data simulator takes p as the survival probability for individuals with mean biomarker level. The survival curve for this group is similar to the overall survival curve.".

3. Results section

a. It will be helpful to illustrate the example in the context of motivating case study (reference #8). What is the objective/purpose of design to including enrichment consideration? Why were the parameters of simulation set up at a specific value? What will preliminary data be essential for the simulation to be useful?

Response: Please note that Example 1 does not illustrate the BioPETsurv data simulator. Because of proprietary data issues we were not able to analyze the data from the actual study and instead generated (outside of BioPET) synthetic data similar to the real data.

b. Figures 1 and 2, it is helpful to show the pattern change by the level of enrichment, but the clinical trial sample size and the number of patients screened could be too high to be realistic.

Response: We agree there is no guarantee a prognostic enrichment biomarker will make trial sample size and number of patients to screen realistic. This is precisely why it is so important to assess a biomarker for its capacity to be useful for prognostic enrichment. In Example 1, it is conceivable that a 7,000 patient trial (36-month trial with no enrichment) would be unrealistic but 

a 4,000 patient trial (36-month trial with 75% enrichment) would be realistic if enough patients were available for such a high level of enrichment. 

c. In a real trial design, how to reasonably determine those parameters for simulation?

Response: In developing BioPETsurv, we carefully considered the simulation parameters and believe these are parameters investigators will be comfortable specifying for the disease areas in which they specialize. For example, in cancer 5-year survival rates are routinely reported, which naturally translates to a survival rate and a time horizon to specify for the simulator. 

d. I hope these tutorial examples can help future users to design their next trial.

Response: Thank you.

4. Discussion Section

a. What is the novelty of the proposed study? Under what circumstance, the end-users should consider using the proposed approach.

Response: We have revised the Discussion extensively to more clearly describe the circumstances in which BioPETsurv will be useful.

b. Any other similar studies, if yes, what are the advantages and disadvantages of the current one.

Response: We have not been able to identify any other publications (other than the original BioPET paper) that presents a similar, comprehensive framework for prognostic enrichment biomarkers.

c. Any other considerations the end-users should be aware of in order to generate the simulation that closely mimics the reality.

Response: We are uncertain what the reviewer has in mind here. We believe the simulation inputs and the BioPET outputs are adequately described for the audience of this paper. In addition, we include a paragraph in the Discussion that summarizes the required inputs and the assumptions of the simulator.

---

## [Decision Letter · Decision Letter 1]

18 Aug 2020

PONE-D-20-12411R1

BioPETsurv: Methodology and open source software to evaluate biomarkers for prognostic enrichment of time-to-event clinical trials

PLOS ONE

Dear Dr. Parikh,

Thank you for submitting your manuscript to PLOS ONE. After careful consideration, we feel that it has merit but does not fully meet PLOS ONE’s publication criteria as it currently stands. Therefore, we invite you to submit a revised version of the manuscript that addresses the points raised during the review process.

Please consider adding illustrations in supplement about additional situation where the Reduction in Total Cost is not a monotonic function of the Screening Threshold.

We look forward to receiving your revised manuscript.

Kind regards,

Qin Liu

Academic Editor

PLOS ONE

Reviewers' comments:

Reviewer's Responses to Questions

**Comments to the Author**

1. If the authors have adequately addressed your comments raised in a previous round of review and you feel that this manuscript is now acceptable for publication, you may indicate that here to bypass the “Comments to the Author” section, enter your conflict of interest statement in the “Confidential to Editor” section, and submit your "Accept" recommendation.

Reviewer #1: (No Response)

Reviewer #2: All comments have been addressed

2. Is the manuscript technically sound, and do the data support the conclusions?

Reviewer #1: Yes

Reviewer #2: Yes

3. Has the statistical analysis been performed appropriately and rigorously? 

Reviewer #1: Yes

Reviewer #2: Yes

4. Have the authors made all data underlying the findings in their manuscript fully available?

Reviewer #1: Yes

Reviewer #2: Yes

5. Is the manuscript presented in an intelligible fashion and written in standard English?

Reviewer #1: Yes

Reviewer #2: Yes

6. Review Comments to the Author

Reviewer #1: Please see attached.

Limit 100 to 20000 Characters

Limit 100 to 20000 Characters

Limit 100 to 20000 Characters

Reviewer #2: (No Response)

7. PLOS authors have the option to publish the peer review history of their article (what does this mean?). If published, this will include your full peer review and any attached files.

Reviewer #1: **Yes: **Jiangtao Gou

Reviewer #2: No

---

## [Author Response · Author response to Decision Letter 1]

2 Sep 2020

Reviewers' comments

Reviewer #1 and #2 agreed that our article is technically sound piece of research with data that support the conclusions, and that our analysis is appropriate and rigorous. Reviewer #1 had one additional comment:

Reviewer #1: Authors have edited this manuscript successfully. I only have one comment.

Authors have added additional descriptions in Discussion for the previous comment: “Under the settings in Table 1 and 2, the Reduction in Total Cost is generally an increasing function of the Screening Threshold. Authors can add additional situation where the Reduction in Total Cost is not a monotonic function of the Screening Threshold. For example, with effect size 2, Cost of screening a patient to determine trial eligibility 100 and Cost of running a patient through the trial 100, the maximum Reduction in Total Cost is achieved around Screening Threshold = 75%.” In order to keep the manuscript succinct, authors decided not to include additional illustrations. However, readers will be benefited from these additional illustrations. Therefore, authors may include these part into supplement material.

Response: Our revised submission includes Supplementary Tables S1 and S2 and Supplementary Figures S1 and S2 to illustrate these additional situations.

---

## [Decision Letter · Decision Letter 2]

8 Sep 2020

BioPETsurv: Methodology and open source software to evaluate biomarkers for prognostic enrichment of time-to-event clinical trials

PONE-D-20-12411R2

Dear Dr. Parikh,

We’re pleased to inform you that your manuscript has been judged scientifically suitable for publication and will be formally accepted for publication once it meets all outstanding technical requirements.

Kind regards,

Qin Liu

Academic Editor

PLOS ONE

Additional Editor Comments (optional):

Reviewers' comments:

Reviewer's Responses to Questions

**Comments to the Author**

1. If the authors have adequately addressed your comments raised in a previous round of review and you feel that this manuscript is now acceptable for publication, you may indicate that here to bypass the “Comments to the Author” section, enter your conflict of interest statement in the “Confidential to Editor” section, and submit your "Accept" recommendation.

Reviewer #1: All comments have been addressed

2. Is the manuscript technically sound, and do the data support the conclusions?

Reviewer #1: Yes

3. Has the statistical analysis been performed appropriately and rigorously? 

Reviewer #1: Yes

4. Have the authors made all data underlying the findings in their manuscript fully available?

Reviewer #1: Yes

5. Is the manuscript presented in an intelligible fashion and written in standard English?

Reviewer #1: Yes

6. Review Comments to the Author

Reviewer #1: Please see attached.

Please see attached.

Please see attached.

Please see attached.

Please see attached.

7. PLOS authors have the option to publish the peer review history of their article (what does this mean?). If published, this will include your full peer review and any attached files.

Reviewer #1: **Yes: **Jiangtao Gou

---

## [Editor Report · Acceptance letter]

10 Sep 2020

PONE-D-20-12411R2 

BioPETsurv:  Methodology and open source software to evaluate biomarkers for prognostic enrichment of time-to-event clinical trials 

Dear Dr. Parikh:

I'm pleased to inform you that your manuscript has been deemed suitable for publication in PLOS ONE. Congratulations! Your manuscript is now with our production department. 

Kind regards, 

on behalf of

Dr. Qin Liu 

Academic Editor

PLOS ONE